# Detecting Oropharyngeal and Esophageal Emptying by Submental Ultrasonography and High-Resolution Impedance Manometry: Intubated vs. Non-Intubated Video-Assisted Thoracoscopic Surgery

**DOI:** 10.3390/diagnostics10121079

**Published:** 2020-12-12

**Authors:** Chih-Jun Lai, Jin-Shing Chen, Shih-I Ho, Zhi-Yin Lu, Yi-Ju Huang, Ya-Jung Cheng

**Affiliations:** 1Institute of Epidemiology and Preventive Medicine, National Taiwan University, Taipei 100025, Taiwan; littlecherrytw@gmail.com; 2Department of Anesthesiology, National Taiwan University Hospital, National Taiwan University College of Medicine, Taipei 100225, Taiwan; szu.i.ho@gmail.com (S.-IH.); coma_lu@hotmail.com (Z.-Y.L.); kimeerica@hotmail.com (Y.-J.H.); 3Department of Surgery, National Taiwan University Cancer Center, National Taiwan University College of Medicine, Taipei 106037, Taiwan; chenjs@ntu.edu.tw; 4Department of Anesthesiology, College of Medicine, National Taiwan University, Taipei 100233, Taiwan; 5Department of Anesthesiology, National Taiwan University Cancer Center, Taipei 106037, Taiwan

**Keywords:** anesthesia, endotracheal intubation, esophageal emptying, high-resolution impedance manometry, oropharyngeal emptying, swallowing, video-assisted thoracosopic surgery

## Abstract

Postoperative swallowing, affected by general anesthesia and intubation, plays an important part in airway and oral intake safety regarding effective oropharyngeal and esophageal emptying. However, objective evidence is limited. This study aimed to determine the time required from emergence to effective oropharyngeal and esophageal emptying in patients undergoing non-intubated (N) or tracheal-intubated (I) video-assisted thoracoscopic surgery (VATS). Hyoid bone displacement (HBD) by submental ultrasonography and high-resolution impedance manometry (HRIM) measurements were used to assess oropharyngeal and esophageal emptying. HRIM was performed every 10 min after emergence, up to 10 times. The primary outcome was to determine whether intubation affects the time required from effective oropharyngeal to esophageal emptying. The secondary outcome was to verify if HBD is comparable to preoperative data indicating effective oropharyngeal emptying. Thirty-two patients suitable for non-intubated VATS were recruited. Our results showed that comparable HBDs were achieved in all patients after emergence. Effective esophageal emptying was achieved at the first HRIM measurement in 11 N group patients and 2 I group patients (*p* = 0.002) and was achieved in all N (100%) and 13 I group patients (81%) within 100 min (*p* = 0.23). HBD and HRIM are warranted for detecting postoperative oropharyngeal and esophageal emptying.

## 1. Introduction

Successful postoperative swallowing includes efficient oropharyngeal and esophageal emptying [1]. Efficient oropharyngeal emptying is crucial for postoperative airway safety. With adequate muscle power to propel the bolus from the oropharynx into the upper esophageal sphincter (UES) [2], oropharyngeal emptying keeps the airway clear, preventing choking and aspiration in advance of inducing the cough reflex. Efficient esophageal peristalsis and emptying, i.e., propelling the bolus through the esophagus and lower esophageal sphincter into the stomach [3], are essential for smooth postoperative oral intake. The possibility of postoperative regurgitation and even aspiration risk may also increase until effective esophageal peristalsis and emptying functions are regained [4].

General anesthesia and tracheal intubation may interfere with postoperative swallowing [5]. However, the effects of intubation during operations and anesthesia on oropharyngeal and esophageal emptying have seldom been investigated. Most previous studies have focused on the effects of prolonged intubation (>48 h) through questionnaires or flexible endoscopic evaluation of swallowing [6,7,8,9,10].

In this study, muscle power and fluid passage via sequential pressure transmission were used to investigate both oropharyngeal and esophageal emptying. The gold standard diagnostic for oropharyngeal dysphagia is video fluoroscopy [11]. The spatial and temporal measurement of hyoid bone displacement (HBD) during swallowing has been widely used [12,13]. However, radiation exposure is still a concern. HBD measurement via submental ultrasonography, a non-invasive method without radiation exposure, was used to measure oropharyngeal muscle power. The accuracy and reliability of ultrasonography have been proven in comparisons with video fluoroscopy [14]. High-resolution impedance manometry (HRIM) with multiple manometric sensors and impedance channels is used to detect fluid boluses through low-impedance signaling in concordance with esophageal peristalsis [15,16,17]. HRIM is a powerful tool that can confirm whether successful oropharyngeal and esophageal emptying occur, and can determine the mechanisms of failure in cases of unsuccessful emptying [18,19].

In the present study, we aimed to determine the time required to regain successful oropharyngeal and esophageal emptying through submental ultrasonography and HRIM for non-intubated (N) or tracheal-intubated (I) patients after video-assisted thoracoscopic surgery (VATS). As in clinical practice, complete emergence is believed to be the time when patients have the ability to protect their airways, so we measured HBD and started the first HRIM measurement immediately after emergence. The primary outcome was to determine whether tracheal intubation affects postoperative swallowing. The time required from effective oropharyngeal to esophageal emptying was compared between the non-intubated and intubated groups. The secondary outcome was to verify if HBDs measured after emergence and compared to preoperative data could indicate effective oropharyngeal emptying. The feasibility of non-invasively measuring HBDs using submental ultrasonography to detect postoperative oropharyngeal emptying was also determined.

## 2. Materials and Methods

### 2.1. Study Participants

This study protocol (ClinicalTrials. gov. registration number NCT03711461) was approved by the Research Ethics Committee of National Taiwan University Hospital, Taipei, Taiwan (no. 201804093RIND, date of approval: 2 July 2018). The trial was conducted at the Department of Anesthesiology at the same institution from November 2018 to December 2019. Thirty-two patients were recruited to participate in this randomized controlled trial. All participants provided verbal and written consent. We recruited patients who were candidates for non-intubated VATS. The criteria for this procedure were as follows: tumors smaller than 6 cm; peripheral lesions; no evidence of severe adhesion; and no evidence of chest wall, diaphragm, or main bronchus involvement. Exclusion criteria for this procedure were potential airway complications such as bronchial tumors or hemothorax, class II or higher obesity [20], coagulopathy, or an American Society of Anesthesiologists Classification Physical Status Classification (ASA) greater than III. Additional exclusion criteria for the study were as follows: (1) gastroesophageal reflux disease (GERD) symptoms or diagnosis via panendoscopy; (2) medication involving any antacid or prokinetic drugs; (3) any previous gastrointestinal or abdominal surgeries.

### 2.2. Protocol

Sequentially numbered, sealed, opaque envelopes were used to randomize treatment assignment after patient enrolment and before the protocol commenced. Treatment assignment was anonymized following randomization. The preoperative fasting time followed the guidelines for enhanced recovery after lung surgery [21]. Clear fluids were allowed up until 2 h before the induction of anesthesia and solids up until 6 h before the induction of anesthesia. We measured HBD using submental ultrasonography before the surgery using the handheld method [22,23]. The patients were monitored using pulse oximeters and electrocardiography results as well as blood pressure, end-tidal carbon dioxide concentration, and frontal bispectral index (BIS) values. The BIS values were assessed using a bispectral index (BIS) monitor (BIS Quatro, Aspect Medical System, Norwood, MA, USA). The BIS monitor is a quantitative electroencephalographic device that is widely used to assess the hypnotic component of anesthesia [24]. Anesthesia was induced and maintained through target-controlled infusion of intravenous propofol and remifentanil (TCI, Injectomat TIVA Agilia, Fresenius Kabi GmbH, Graz, Austria) to maintain a BIS level between 40 and 60, which reflects a level of unconsciousness suitable for surgery [24]. An arterial catheter was inserted for hemodynamic monitoring and arterial blood gas analysis.

In the non-intubated (N) group, patients were preoxygenated with Transnasal Humidified Rapid-Insufflation Ventilatory (THRIVE, Fisher and Paykel Healthcare, Auckland, New Zealand) at an initial flow of 20 L/min before anesthesia. Oxygen flow was temporarily suspended immediately before iatrogenic pneumothorax and was resumed up to 10 L/min after the surgeon confirmed complete lung collapse using thoracoscopic observation [25,26]. The end-tidal carbon dioxide was measured by insertion of a detector into one nostril, which also helped us to monitor the patients’ respiratory rates [26]. The VATS procedure in both groups was performed as previously described with a thoracoscopic intercostal nerve block, produced by infiltration of 0.5% bupivacaine (1.5 mL for each intercostal space) from the third to the eighth intercostal nerve after the first thoracoscopic port [27]. The infusion of propofol and remifentanil was adjusted intraoperatively to maintain BIS levels between 40 and 60. In the N group, the goal was a respiratory rate between 12 and 18 breaths/min to ensure a smooth operation field. If the respiratory rate could not be maintained within 12–18 breaths/min or if an initially non-intubated VATS operation required subsequent intubation, the patients were excluded from the analysis.

In the I group, 1 mg/kg rocuronium was administered after induction to facilitate endotracheal tube insertion with train-of-four (TOF, TOF-Watch^®^ S, Organon, Oss, The Netherland) monitors. The patients were intubated with a single-lumen endotracheal tube (ST-ETT) and bronchial blocker (BB, Coopdech Endobronchial Blocker Tube, Daiken Medical Corp, Osaka, Japan). The cuff pressure of the ST-ETT was not allowed to exceed 30 cm H_2_O as measured by a pocket cuff pressure gauge [28]. After tracheal intubation, all BBs were placed via the ST-ETT. A flexible bronchoscope was used to check the BB positioning, which should be positioned distally in the main bronchus at the surgical side for lung collapse [29]. All intubated patients received volume-controlled ventilation using an anesthesia machine (Aisys CS^2^, GE, USA). Parameters before one lung ventilation (OLV) were as follows: tidal volume, 8–10 mL/kg; respiratory rate, 12–18 breaths/min to maintain the arterial carbon dioxide tension at 35 to 45 mm Hg and ETCO_2_ between 30 to 35 mm Hg; inspiratory–expiratory ratio, 1:2; fraction of inspired oxygen (FiO_2_), 1.0; oxygen flow, 1 L/min. The peak airway was kept below 20 cm H_2_O. During OLV, the tidal volume was 4–5 mL/kg. The peak pressure was kept below 30 cm H_2_O. The sugammadex administration followed the recommendation according to the response to TOF stimulation [30]. When the reversal of the displayed TOF ratio achieved a value greater than 1, the patients were extubated [31].

In both groups, an HRIM catheter was inserted after surgery and before emergence. All patients were sent to the post-anesthetic care unit (PACU) after complete emergence with BIS > 85, which reflects being awake [32]. The ability to actively cough and extend the tongue outside the mouth were tested immediately after complete emergence. Postoperative HBD was immediately measured using submental ultrasonography [22,23]. The first HRIM measurement was performed while the patients swallowed 10 mL of normal saline, 10 min after emergence, and HRIM measurements were repeated every 10 min up to 10 times for each patient.

Postoperative records, including those concerning time to oral intake, hypoxia, and re-intubation, were retrospectively collected through follow-up in the general ward.

### 2.3. Equipment

#### 2.3.1. High-Resolution Impedance Manometry

Manometric studies were conducted using an HRIM catheter with a 10 Fr outer diameter and solid-state assembly of 36 circumferential pressure sensors at 1 cm intervals and 12 impedance segments at 2 cm intervals (MMS, Enschede, The Netherlands). Before each recording, the catheter was calibrated according to the manufacturer’s instructions. Distal impedance and pressure signals were positioned within the hiatus after transnasal placement of the HRIM, and the lower esophageal sphincter (LES) was identified via crural diaphragm contraction. HRIM measurements were initiated after confirmation of these positions.

#### 2.3.2. Measurements

Assessing successful oropharyngeal emptying using hyoid bone displacement measured by submental ultrasonography

Ultrasonography was performed using a 2–5 MHz curvilinear array transducer and a Sonosite X Porte system (Fujifilm, Sonosite, Bothell, WA, USA) with image compounding technologies. Submental ultrasonography was used to measure HBD (cm) before and during swallowing [14,23]. The HBD measurement was taken while 10 mL of water was being swallowed, and this measurement was repeated three times. The average of the three measurements taken was calculated. Successful oropharyngeal emptying was defined as postoperative HBD returning to the range of preoperative data.

Assessing successful oropharyngeal and esophageal emptying using HRIM data

The acquisition system allowed the export of raw pressure and impedance data from the HRIM assembly to a spreadsheet template (Microsoft Excel; Microsoft Corporation, Redmond, WA, USA). The HRIM data for each patients were exported to MATLAB (MathWorks Inc., Natick, MA, USA) for pressure flow analysis. To demonstrate successful esophageal emptying, bolus transmission (BT) through the esophagus and LES into the stomach was assessed using pressure and flow analysis. Successful esophageal emptying was defined as fluid passage through the lower esophageal sphincter into the stomach. Bolus transmission through the esophagus and successful bolus transmission through the LES into the stomach were analyzed separately.

(1)Bolus transmission through the esophagus:

Successful bolus transmission from the UES, esophagus, and LES into the stomach is shown in Figure 1. Transmission through the portion of the esophagus was determined as the impedance-measuring segment of the time interval between bolus entry (50% decrease from baseline relative to nadir) and bolus exit (recovery of 50% of baseline; Figure 2) [33].

(2)Successful bolus transmission through the LES into the stomach

Successful BT through the LES was defined as both bolus presence and a flow-permissive pressure gradient. The onset of bolus presence was defined as the drop in impedance to 90% of the nadir, while bolus passage was defined as a return of impedance to 50% of the baseline. A pressure gradient was considered permissive when the esophageal pressure was higher than both the LES and the intra-gastric pressure signals (Figure 3) [16,34,35].

### 2.4. Statistical Analysis

Because of the limited data about the time required to regain effective postoperative esophageal and oropharyngeal emptying after emergence, we took the postoperative complication incidence from previous studies of intubated and non-intubated VATS as the references to calculate the required sample size. To estimate the complications for the non-intubated group, we used data from our non-intubated VATS data set, which spanned seven years. These data were published in 2019 by Hung et al. [36]. To estimate the complications for the intubated group, we used the study by Russo et al. as a reference [37]. Therefore, these values were set at 0% in the N group, referencing Hung et al., and at 50% in the I group, referencing Russo et al. [36,37]. Using MedCalc statistical software version 18.10.2 (MedCalc Software, Ostend, Belgium), we estimated that a sample size of 15 patients per group would provide 80% power with an alpha level of 0.05. We therefore decided to enroll 15 patients per group. After we started this study and assessed the first eight patients, we reconfirmed this sample size calculation.

The Shapiro–Wilk test was used to determine the normal distribution for continuous variables, and the results are reported as mean (standard deviation) and median [interquartile range] values. Independent *t*-tests or Mann–Whitney *U* tests and unpaired *t*-tests or Wilcoxon tests were used for univariate analysis of normally and non-normally distributed continuous variables, respectively. Fisher’s exact test was used for categorical variables.

*p*-values less than 0.05 were considered significant. All statistical analyses were performed using SPSS software, version 22 (IBM Corp., Armonk, NY, USA).

## 3. Results

Thirty-two patients completed the study protocol (Figure 4). In the I group, all patients were allowed to be transferred to the PACU after achieving a TOF ratio of >1. No patients in the N group were excluded due to an inability to maintain a respiratory rate within 12–18 breaths/min or requirement for tracheal intubation. All patients in both groups were sent to the PACU after restoration of BIS to the value of at least 85. We did not experience any episode of hypoxia or reintubation in our study. There were no significant differences between the N and I groups in terms of their demographic data or surgical procedures (Table 1 and Table 2).

Successful oropharyngeal emptying assessed via HBD submental ultrasonography after complete emergence

Immediately after emergence with BIS > 85, all patients from both groups were able to cough voluntarily. The preoperative HBD data were comparable between the N and I groups (N: 1.67 ± 0.14 cm; I: 1.66 ± 0.14 cm, *p* = 0.82). Postoperative HBD values comparably returned to preoperative levels immediately after emergence in both the N (1.67 ± 0.12 cm, *p* = 0.46) and I (1.64 ± 0.13 cm, *p* = 0.30) groups.

Success of oropharyngeal emptying and esophageal emptying assessed based on bolus transmission through the esophagus, LES and into the stomach, measured with HRIM

A total of 319 esophageal swallows were analyzed (N group: 159 swallows; I group: 160 swallows). Only one patient in the N group failed to complete the last swallow. Successful BT passing through the UES into esophagus occurred in all patients from the first measurement.

In assessing the BT through the LES into the stomach, 11/16 patients exhibited successful BT at the first measurement in the N group, compared with 2/16 patients in the I group (*p* = 0.002; Figure 4). The remaining five patients in the N group experienced successful BT into the stomach within 80 min. In the I group, 13/16 patients (82%) experienced successful BT within 100 min (Figure 5).

We observed two types of failed esophageal emptying: fluid stasis above the LES due to an inability to relax the LES (achalasia-like), and insufficient distal esophageal contraction to push a fluid bolus through the LES. BT failure occurred in 81/159 and 97/160 swallows in the N and I groups, respectively.

Postoperative oral intake commenced within five hours of surgery and did not differ significantly between the two groups (Table 2).

## 4. Discussion

Our results confirmed that effective oropharyngeal emptying is achieved immediately after complete emergence from general anesthesia, as proven by submental ultrasonography and high-resolution impedance manometry (HRIM). This is the first study to apply submental ultrasonography and HRIM to assess effective postoperative swallowing. Submental ultrasonography has been proven to have good accuracy for HBD measurement when compared with video fluoroscopy [23,38]. By using this non-invasive method, our results confirmed that endotracheal intubation does not affect oropharyngeal emptying without residual muscle relaxation. Given that the HBD values for the patients in this study were comparable to those of the normal population [39], endotracheal intubation did not delay oropharyngeal emptying. The clearing effect of oropharyngeal emptying by the fluid bolus through the UES opening into the esophagus were verified by the HRIM measurements. Our results showed that the patients were able to prevent fluid retention around the larynx to avoid choking and subsequent coughing. We confirmed the feasibility of applying these standards in the PACU and considered it safe for patients to sip water after emergence to relieve throat dryness.

This is also the first study to show that esophageal emptying may be delayed by general anesthesia or with tracheal intubation in spite of successful oropharyngeal emptying. We found that intubation resulted in a longer time until effective esophageal emptying was achieved. In our opinion, the cuff pressure of the endotracheal tube may affect upper esophageal motility [40,41], although the cuff pressure did not exceed 30 H_2_O in any of our patients. The upper and lower portions of the esophagus influence each other [1]. This may be the cause of the delayed recovery of esophageal emptying. However, successful esophageal emptying was achieved within 100 min in most patients (81%) undergoing VATS, and the time to oral intake was similar in both groups. In clinical practice, the time to oral intake is affected by multiple factors such as the ward routine, postoperative gastrointestinal discomfort, and appetite. However, our results showed that esophageal emptying can be regained within hours. We also demonstrated that relative to other measurement techniques, HRIM is a powerful tool for assessing esophageal emptying, with acceptable levels of discomfort caused by the retained catheter in the throat. Although traditional manometry was used to measure LES and UES resting and relaxation pressures in patients receiving lung resection in a previous study, bolus flow was not assessed in that study [42]. In addition, while the barium swallow test can be used to detect BT through the esophagus into the stomach, pressure propagation cannot be measured simultaneously [42]. Further, repeated measurements involving exposure to radioactive barium to determine the time of esophageal emptying have undesirable side effects.

Our study also uses HRIM to demonstrate two mechanisms of failed postoperative esophageal emptying after anesthesia and operations. In the first type of failure, the fluids reached the distal esophagus, where in the absence of LES relaxation they were then retained above the stomach. This seems similar to what occurs in patients diagnosed with achalasia [43]. In the second type of failure, the muscle power of the esophageal peristalsis was insufficient to push a fluid bolus through the barrier of the LES into the stomach. This was similar to the HRIM criteria for the Chicago classification of ineffective esophageal motility disorder [44,45]. However, in our opinion, using these criteria for diagnosis was not appropriate for our patients, since they did not actually have esophageal motility problems [46]. It is thus better to use the raw HRIM data merely to describe successful or failed esophageal emptying. In addition, postoperative oral intake should only be started after successful esophageal emptying has resumed. Prior to this, the fluid retained above the LES would increase regurgitation, which may increase the risk of aspiration [47]. Despite the delay in esophageal emptying in the intubated patients, all our patients had a faster return to oral intake (N group, 4.11 h; I group, 3.35 h) than in a previous study (6.5 and 13.8 h, respectively) [48].

In our study, the neuromuscular blockade (NMB) rocuronium was used during the operations. To avoid the residual effects of the NMB, we used TOF monitors to confirm complete recovery from NMB. The criterion was a TOF ratio greater than 1.0 [49]. Sugammadex, which selectively binds rocuronium and reverses its NMB action [50], does not have the same cholinergic effects as traditional anti-acetylcholinesterase [51], which has been reported to affect esophageal tone [52].

Our study had some limitations. First, we did not apply HRIM preoperatively. We did not want to measure preoperative esophageal emptying because the associated anxiety and stress may have affected the esophageal peristalsis [53]. Since we had excluded patients with GERD or other gastrointestinal problems, most patients regained esophageal emptying within 100 min. Second, we concluded that the three patients in the I group did not complete a successful BT into the stomach within 100 min based on retrospective analysis of the raw data and not the processed data from the commercialized software package used for monitoring esophageal emptying. The actual time taken until successful esophageal emptying occurred remains unknown. However, the clinical impacts appear to have been limited, since the actual time to oral intake in the two groups was similar. Third, our study’s potential generalizability to clinical practice is limited due to the small sample size. Our results indicate that for the assessment of upper gastrointestinal tract function, measurement of perioperative HRIM is feasible and can be used to differentiate causes of failure. Fourth, our study used patients suitable for non-intubated VATS for both non-intubated and tracheal-intubated groups. Different respiratory patterns (spontaneous vs. controlled) may affect the postoperative swallowing. However, our results showed that complete emergence and the complete recovery of neuromuscular function ensure airway safety with effective oropharyngeal emptying.

## 5. Conclusions

The use of submental ultrasonography and HRIM is warranted for detection of postoperative oropharyngeal and esophageal emptying. Although effective oropharyngeal emptying without choking was proven to be achievable with complete emergence and complete recovery of neuromuscular function, anesthesia and tracheal intubation may delay effective esophageal emptying. Submental ultrasonography could be used in all surgeries that may affect patients’ oropharyngeal swallowing and for all patients with high aspiration risk, such as for older adults or stroke patients. It also helps to evaluate the rehabilitation of oropharyngeal swallowing after surgery. HRIM could be used in patients undergoing all surgeries associated with affected esophageal dysfunction, such as laparoscopic esophagectomy. HRIM could serve as a powerful tool for differential diagnosis of oral intake difficulties and for postoperative follow-up.

## Figures and Tables

**Figure 1 diagnostics-10-01079-f001:**
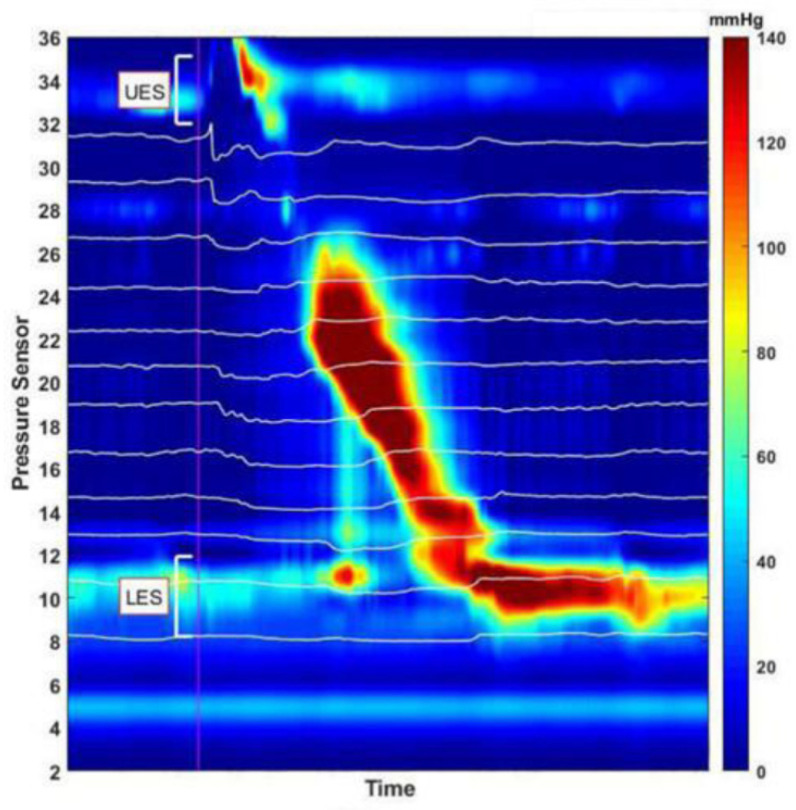
Assessment of successful esophageal emptying using high-resolution impedance manometry (HRIM) monitoring. Data from pressure sensors on the HRIM catheter are displayed on the y-axis and the time is displayed on the x-axis. The graph indicates the esophageal pressure topography, from the pharynx to the stomach, showing two high-pressure zones. The upper high-pressure zone is the upper esophagus sphincter (UES), while the lower zone is the lower esophageal sphincter (LES). The impedance of the 12 sequential channels from the esophagus into the stomach is represented as 12 white horizontal lines. Effective esophageal contraction relies on bolus transmission, during which the level of impedance decreases.

**Figure 2 diagnostics-10-01079-f002:**
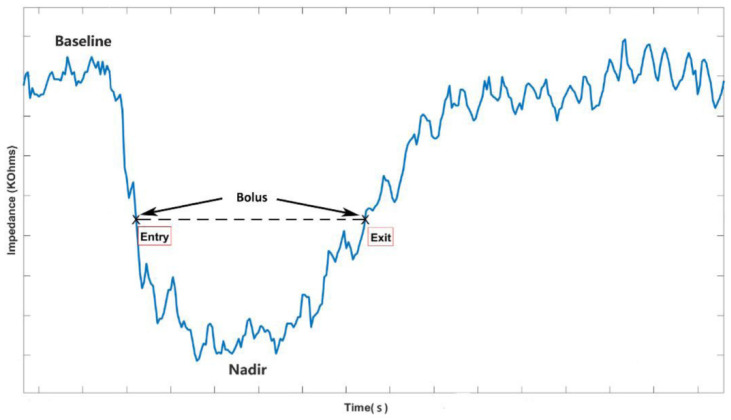
Successful bolus transmission through one impedance channel in the esophagus. Impedance changes observed during bolus transmission between a single pair of measurement rings separated by 2 cm. The impedance signal dropped when the fluid bolus material passed the impedance-measuring segment. Bolus entry occurred when impedance dropped by 50% from baseline relative to nadir, and bolus exit occurred when it recovered by 50% from nadir to baseline. The black dotted lines indicated the bolus from entry to exit.

**Figure 3 diagnostics-10-01079-f003:**
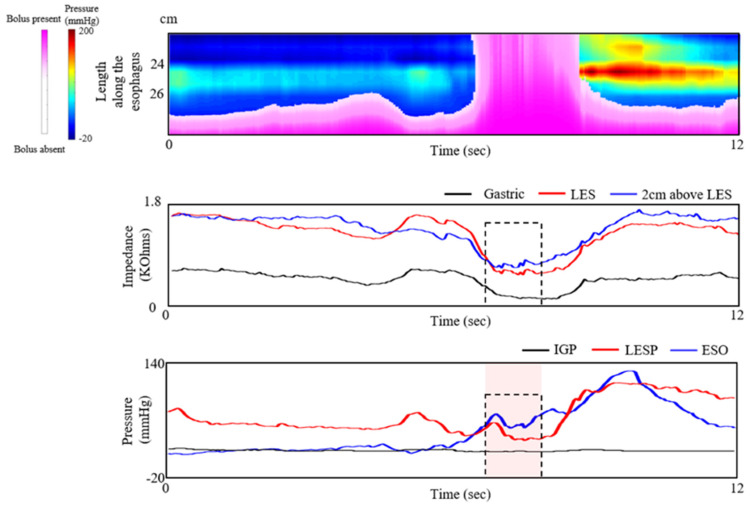
Successful esophageal emptying indicated by successful bolus transmission (BT) through the lower esophageal sphincter (LES). Upper panel: The esophageal pressure topography of the distal esophagus and LES. Middle panel: Impedance signals used to determine the bolus presence. Lower panel: Pressure signals used to identify the periods during which a flow-permissive gradient existed, when esophageal pressure (blue line) exceeded both LES and intragastric pressures. BT occurred when both criteria (bolus presence and trans-LES flow permissive pressure gradient) were met.

**Figure 4 diagnostics-10-01079-f004:**
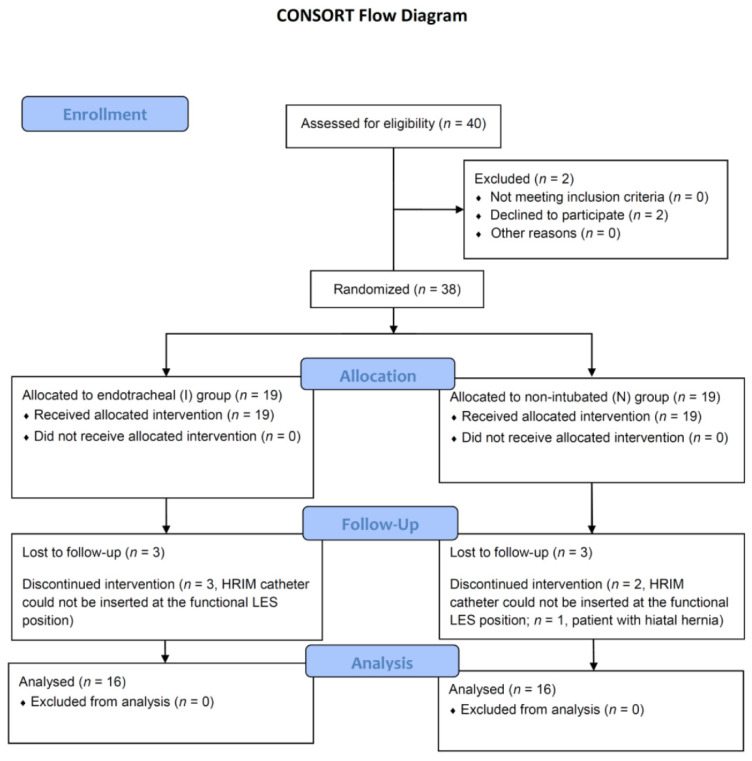
CONSORT flowchart of patient enrollment. N: non-intubated video-assisted thoracoscopic surgery group; I: intubated video-assisted thoracoscopic surgery group; LES: lower esophageal sphincter; HRIM: high-resolution impedance manometry.

**Figure 5 diagnostics-10-01079-f005:**
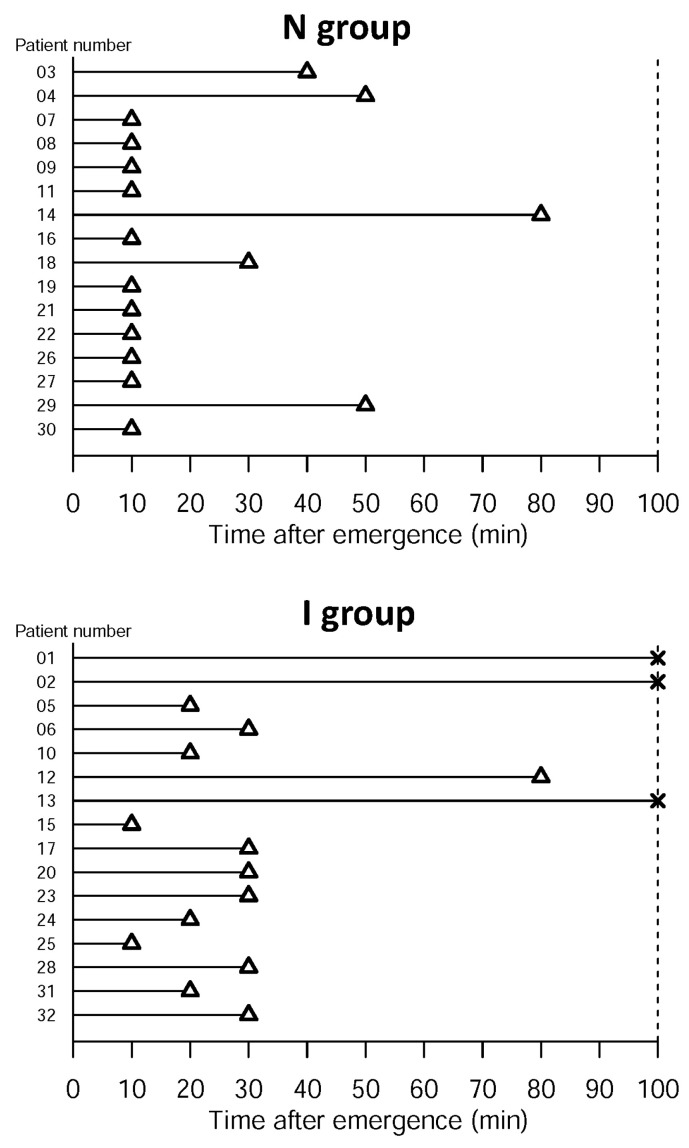
Time to successful esophageal emptying after emergence for both groups. Time to the first presentation of successful bolus transmission (BT) through the lower esophageal sphincter based on individual high-resolution impedance manometry (HRIM) measurements. N: non-intubated video-assisted thoracoscopic surgery group; I: intubated video-assisted thoracoscopic surgery group. Measurement: HRIM was performed every 10 min, up to 10 times for each patient. Note: Δ: the first successful BT; ╳: the patient presented no successful BT within 10 measurements.

**Table 1 diagnostics-10-01079-t001:** Characteristics of patients in N and I groups.

Variables	N Group (*n* = 16)	I Group (*n* = 16)	*p* Value
Female, *n* (%)	11(68.75)	10 (67.50)	0.71
Age, y	56.00 (4.71)	56.38 (6.13)	0.85
Body weight, kg	57.09 (8.30)	62.24 (10.81)	0.14
Body height, cm	161.17 (8.20)	161.63 (9.64)	0.89
ASA I/II/III, *n*	7/9/0	3/12/1	0.25
Smoking (yes/no), *n*	2/14	2/14	1
Pulmonary function test, % of prediction			
FVC	111.99 (13.95)	107.62 (16.43)	0.42
FEV1	108.75 (16.79)	106.82 (16.50)	0.75
Comorbidity, *n*			
COPD	0	0	
Asthma	1	0	1
Cardiac disease	0	0	
Hypertension	1	2	0.6
Diabetes mellitus	0	1	1

Abbreations: ASA: American Society of Anesthesiologist Physical Status Classification; COPD: chronic obstructive pulmonary disease; FEV1: forced expiratory volume in 1 s; FVC: forced vital capacity; N: non-intubated video-assisted thoracoscopic surgery; I: intubated video-assisted thoracoscopic surgery. Data are presented as mean (standard deviation) or numbers.

**Table 2 diagnostics-10-01079-t002:** Types of surgical procedures in patients receiving N and I groups.

Surgical Method	N Group (*n* = 16)	I Group (*n* = 16)	*p* Value
Wedge resection, *n* (%)	9 (76.62)	9 (70.35)	1
Segmentectomy, *n* (%)	2 (5.95)	2 (6.98)	1
Lobectomy, *n* (%)	5 (19.05)	5 (19.19)	1
Operation time, min	70.50 (60.00–95.50)	72.00 (49.50–134.50)	0.85
Anesthetic time, min	106.00 (86.00–145.00)	120.00 (77.00–164.00)	0.95
Blood loss			0.6
<50 c.c., *n* (%)	14 (87.50)	13 (81.25)	
50 to 150 c.c., *n* (%)	1 (6.25)	3 (18.75)	
>150 c.c., *n* (%)	1 (6.25)	0 (0.00)	
Blood transfusion	0	0	
Intraoperative SpO2 < 90	0	0	
Postoperative time to oral intake, hours	4.11 (3.25–4.95)	3.35 (2.95–7.29)	0.49
Hospital stay, day	4 (4–6.5)	4.5 (4–5.5)	0.95
Tumor size, mm	10.50 (7.50–17.50)	12.50(5.50–21.50)	0.96

Abbreviations: N: non-intubated video-assisted thoracoscopic surgery; I: intubated video-assisted thoracoscopic surgery. Data are presented as median [interquartile range] or numbers.

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
