# Peer review of "Detecting Oropharyngeal and Esophageal Emptying by Submental Ultrasonography and High-Resolution Impedance Manometry: Intubated vs. Non-Intubated Video-Assisted Thoracoscopic Surgery"

_diagnostics, 2020, doi:10.3390/diagnostics10121079_

Round 1
Reviewer 1 Report
This is an interesting and innovative study:
Major comments:
- Primary outcome and secondary aims should be clearly defined in the abstract (and in the manuscript as well).
- Methods of tracheal intubation is not described appropriately - the readers do not not if a standard endotracheal tube was used or, as often used in thoracic surgery procedures, any method allowing the lung collapse - double-lumen tube or bronchial blocker device.
- It is not clear, how spontaneous ventilation at the rate of 12-18 per minute may be maintained throughout the procedure - what happened if the patient was breathing at the rate of 10 per min or 20 per min - were they excluded from the further analysis?
- Mode of ventilation in the intubated group should be described in detail including peak pressures.
- More information about pilot study serving as a background for power analysis should be provided. Was this pilot performed after the Ethical Approval? How many patients were in that pilot trial and why? Was the study protocol registered with the ClinicalTrials.gov before starting the pilot or afterwards? Minor comments:
- p1L18 - swallowing is not crucial for postoperative recovery - it is rather "one of important factors".
- p2L60 - "In the present study, " or "In this study, "
- p2L67 - the date of the Ethical Approval and name of the Head of the Committee should be provided
- p2L68 - was the trial lasting only one month? - November 2018?
- p2L73 - morbid obesity - this definition does not describe persons with BMI over 35 kg/m2 - it is defined as BMI over 40 kg/m2 or over 35 kg/m2 with other medical problems - e.g. hypertension or diabetes. It could be better to describe that as a "Class II obesity".
- P2L77 - "prokinetic drugs"
- P2L82 - An 8 h fasting period - it is very unusually long, mainly in time when ERAS protocols following the surgery are encouraged elsewhere - the authors should discuss that - was it applicable for a solid food only or also for fluids?
- P2L91 - the authors describe THRIVE as their method of oxygenation during non-intubated VATS. They should provide parameters for the THRIVE - oxygen flow, ETC. I am not completely sure if the term THRIVE is not designated only for patients in apnoe or bradypnoe??? Do the authors mean HFNO - "high flow nasal oxygen"? this should be clarified.
- p2L92 - rocuronium was used to facilitate tracheal intubation in the "I" group. This could influence the difference in recovery of swallowing. The authors should comment this in "Discussion" section.
- p3L94 - "sugammadex was administered after surgery with a TOF ratio 1:1 - what does this mean? - did they want to achieve TOFR of 1.0?
- p3L129 - please explain this sentence - "We divided into two parts to analyze these..."
- p9L228 - "We did not experience any episode of hypoxia and reintubation in our study".
- p9L232 - ...are feasible and first apply...? what the authors do mean?
Author Response
Response to Reviewer 1 Comments
Major comments:
Point 1: Primary outcome and secondary aims should be clearly defined in the abstract (and in the manuscript as well). 

Response 1: Thank you for this valuable suggestions. We have revised the abstract and manuscript body as follows:
- Abstract (Page 1, line 18 to 34, abstract):
“Postoperative swallowing, affected by general anesthesia and intubation, plays an important part on airway and oral-intake safety regarding effective oropharyngeal and esophageal emptying. However, objective evidence is limited. This study aimed to determine the time required from emergence to effective oropharyngeal and esophageal emptying in patients undergoing non-intubated (N) or tracheal-intubated (I) video-assisted thoracoscopic surgery (VATS). Hyoid bone displacement (HBD) by submental ultrasonography and high resolution impedance manometry (HRIM) measurements were used to assess oropharyngeal and esophageal emptying. HRIM was performed every 10 minutes after emergence, up to 10 times. The primary outcome was to determine whether intubation affects the time required from effective oropharyngeal to esophageal emptying. The secondary outcome was to verify if HBDs comparable to preoperative data indicate effective oropharyngeal emptying. Thirty- two patients suitable for non-intubated VATS were assigned to each group. Our results showed that comparable HBDs were achieved in all patients after emergence. Effective esophageal emptying was achieved at the first HRIM measurement in 11 N group patients and two I group patients (p = 0.002); and achieved in all N group (100%) and 13 I group patients (81%) within 100 minutes. HBD and HRIM are warranted for detecting postoperative oropharyngeal and esophageal emptying.”
- Introduction (Page 2, line 68 to 76, introduction):
“As in clinical practice, complete emergence is believed to be the time when patients have the ability to protect their airways, so we measured HBD and started the first HRIM measurement immediately after emergence. The primary outcome was to determine whether tracheal intubation affects postoperative swallowing. The time required from effective oropharyngeal to esophageal emptying was compared between the non-intubated and intubated groups. The secondary outcome was to verify if HBDs measured after emergence and compared to preoperative data could indicate effective oropharyngeal emptying. The feasibility of non-invasively measuring HBDs using submental ultrasonography to detect postoperative oropharyngeal emptying was also determined.”
Point 2: Methods of tracheal intubation is not described appropriately. The readers do not describe if a standard endotracheal tube was used or, as often used in thoracic surgery procedures, any method allowing the lung collapse -double-lumen tube or bronchial blocker device.
Response 2: I thanks for reviewer’s comment. We have added it and reference in the method (Page 3, line 121 to 127, Material and Methods; Reference 27 and 28): “The patients were intubated with a single-lumen endotracheal tube (ST-ETT) and bronchial blocker (BB, Coopdech Endobronchial Blocker Tube, Daiken, Japan). The cuff pressure of the ST-ETT was not allowed to exceed 30 cmH2O as measured by a pocket cuff pressure gauge [27]. After tracheal intubation, all BBs were placed via the SL-ETT. A flexible bronchoscope was used to check the BB positioning, which should be positioned distally in the main bronchus at the surgical side for lung collapse [28].”
Reference:
[27] Lai, CJ.; Liu, CM.; Wu, CY.; Tsai, FF.; Tseng, PH. Fan, SZ. I-Gel is a suitable alternative to endotracheal tubes in the laparoscopic pneumoperitoneum and trendelenburg position. BMC Anesthesiol. 2017, 17, 3.
[28] Bussières, JS.; Somma, J.; Del Castillo, JL.; Lemieux, J.; Conti, M.; Ugalde, PA.; Gagné, N.; Lacasse, Y. Bronchial blocker versus left double-lumen endotracheal tube in video-assisted thoracoscopic surgery: a randomized-controlled trial examining time and quality of lung deflation. Can J Anaesth. 2016, 63, 818-827.
Point 3: It is not clear, how spontaneous ventilation at the rate of 12-18 per minute may be maintained throughout the procedure - what happened if the patient was breathing at the rate of 10 per min or 20 per min - were they excluded from the further analysis?
Response 3: We thank the reviewer for their question. We used the anesthetic depth (BIS 40 to 60) and thoracoscopic intercostal nerve block [1] to maintain the respiratory rate. The thoracoscopic intercostal nerve block was performed in both the intubated and non-intubated groups. We have revised the methods and results as following:
- Methods (Page 3, line 112 to 119, material and methods; Reference 26): The VATS procedure in both groups was performed as previously described with thoracoscopic intercostal nerve block, produced by infiltration of 0.5% bupivacaine (1.5 mL for each intercostal space) from the third to the eighth intercostal nerve after through the first thoracoscopic port [26]. The infusion of propofol and remifentanil was adjusted intraoperatively to maintain BIS levels between 40-60. In N group, the goal was a respiratory rate maintained between 12-18 breaths/min for smooth operation field. If the respiratory rate could not be maintained within 12-18 breaths/min or if an initially non-intubated VATS operation required subsequent intubation, the patients would be excluded from the analysis.
- Results (Page 7, line 231 to 233, Result): No patients in the N group were excluded due to an inability to maintain a respiratory rate within 12-18 breaths /min or requirement for tracheal intubation.
Reference:
[26]. Hung, MH.; Hsu, HH.; Chan, KC.; Chen, KC.; Yie, JC.; Cheng, YJ.; Chen, JS. Non-intubated thoracoscopic surgery using internal intercostal nerve block, vagal block and targeted sedation. Eur J Cardiothorac Surg. 2014, 46, 620-625.
Point 4: Mode of ventilation in the intubated group should be described in detail including peak pressures.
Response 4: We thanks the reviewer for their suggestion. We have revised this section in the Material and Methods:
(Page 3, line 127 to 133, Material and Methods): “All intubated patients received volume- controlled ventilation using an anesthesia machine (Aisys CS2, GE, USA). Parameters before one lung ventilation (OLV) were as follows: tidal volume, 8-10 mL/kg; respiratory rate, 12-18 breaths / min to maintain the arterial carbon dioxide tension at 35 to 45 mm Hg and ETCO2 between 30 to 35 mm Hg; inspiratory/ expiratory ratio, 1:2; fraction of inspired oxygen (FiO2), 1.0; and oxygen flow, 1L/min. The peak airway was kept below 20 cm H2O. During OLV, the tidal volume was 4-5 mL/kg. The peak pressure was kept below 30 cm H2O.”
Point 5: More information about pilot study serving as a background for power analysis should be provided. Was this pilot performed after the Ethical Approval? How many patients were in that pilot trial and why? Was the study protocol registered with the ClinicalTrials.gov before starting the pilot or afterwards?
Response 5: We thanks for reviewer for the valuable comment. We have clarified the sample size calculation in the present version in the method section (Page 6, line 210 to 221, material and methods, reference 34 and 35).
“Because of the limited data about the time required to regain postoperative effective esophageal and oropharyngeal emptying after emergence, we took the postoperative complication incidence from previous studies of intubated and non-intubated VATS as the references to calculate the required sample size. For estimating the complications of the non-intubated group, we used data from our non-intubated VATS data set, which spanned seven years. This data was published in 2019 by Hung et al. [34]. For estimating the complications of the intubated group, we used the Russo et al. study as a reference [35]. Therefore, these values were set at 0% in the N group, referencing Hung et al., and 50% in the I group, referencing Russo et al [34,35]. Using MedCalc statistical software version 19.03, we estimated that a sample size of 15 patients per group would provide 80% power with an alpha level of 0.05. We therefore decided to enroll 15 patients per group. After we started this study and assessed the first eight patients, we reconfirmed this sample size calculation.”
Reference:
[34] Hung, WT.; Hung, MH.; Wang, ML.; Cheng, YJ.; Hsu, HH.; Chen, JS. Nonintubated Thoracoscopic Surgery for Lung Tumor: Seven Years' Experience With 1,025 Patients. Ann Thorac Surg. 2019, 107, 1607-1612.
[35]Russo L;Wiechmann RJ;Magovern JA;Szydlowski GW;Mack MJ.; Naunheim, KS.; Landreneau, RJ. Early chest tube removal after video-assisted thoracoscopic wedge resection of the lung. Ann Thorac Surg. 1998, 66, 1751-1754.
Point 6: p1L18 - swallowing is not crucial for postoperative recovery - it is rather "one of important factors".
Response 6: Many thanks for the reviewer’s suggestion. We have revised the wording as the following (Page 1, line 18 to 20, Abstract): postoperative swallowing, affected by general anesthesia and intubation, plays an important part on airway and oral-intake safety regarding effective oropharyngeal and esophageal emptying.
Point 7: p2L60 - "In the present study, " or "In this study, "
Response 7: We have revised the phrase to “in the present study” (Page 2, line 65, introduction)
Point 8: p2L67 - the date of the Ethical Approval and name of the Head of the Committee should be provided
Response 8: we have added the date of the ethical approval “July 2, 2018.” (Page 2, line 81, Material and Methods)
Point 9: p2L68 - was the trial lasting only one month? - November 2018?
Response 9: we have revised the description for clarity. “The trial was conducted at the Department of Anesthesiology of the same institution from November 2018 to December 2019.” (Page 2, line 81 to 82, Material and Methods)
Point 10: p2L73 - morbid obesity this definition does not describe persons with BMI over 35 kg/m2 - it is defined as BMI over 40 kg/m2 or over 35 kg/m2 with other medical problems - e.g. hypertension or diabetes. It could be better to describe that as a "Class II obesity".
Response 10: We have revised it as the following and also added the reference. “Class II obesity.” (Page 2, line 88, material and methods, reference 20)
Reference:
[20] Catherine Keating, Kathryn Backholer, Emma Gearon, Christopher Stevenson, Boyd Swinburn, Marj Moodie, Rob Carter, Anna Peeters. Prevalence of class-I, class-II and class-III obesity in Australian adults between 1995 and 2011-12. Obes Res Clin Pract. 2015;9(6):553-62.
Point 11: P2L77 - "prokinetic drugs"
Response 11: we have revised this phrase. (Page 2, line 91 to 92, material and methods)
Point 12: P2L82 - An 8 h fasting period - it is very unusually long, mainly in time when ERAS protocols following the surgery are encouraged elsewhere - the authors should discuss that - was it applicable for a solid food only or also for fluids?
Response 12: we have revised the description as the follow:
(Page 3, line 96 to 98, material and methods, reference 21) “The preoperative fasting time was followed the guidelines for enhanced recovery after lung surgery [1]. Clear fluids were allowed up until 2 h before the induction of anesthesia and solids up until 6 h before the induction of anesthesia.”
Reference:
[21] Batchelor, TJP.; Rasburn, NJ.; Abdelnour-Berchtold, E.; Brunelli, A.; Cerfolio, RJ.; Gonzalez, M.; Ljungqvist, O.; Petersen, RH.; Popescu, WM.; Slinger, PD.; Naidu, B. Guidelines for enhanced recovery after lung surgery: recommendations of the Enhanced Recovery After Surgery (ERAS®) Society and the European Society of Thoracic Surgeons (ESTS). Eur J Cardiothorac Surg. 2019, 55, 91-115.
Point 13: P2L91 - the authors describe THRIVE as their method of oxygenation during non-intubated VATS. They should provide parameters for the THRIVE - oxygen flow, ETC. I am not completely sure if the term THRIVE is not designated only for patients in apnoe or bradypnoe??? Do the authors mean HFNO - "high flow nasal oxygen"? this should be clarified.
Response 13: we appreciate the reviewer’s comment. We have answered the reviewer’s question here and have revised the explanation in the Material and Methods.
Answer the question: Transnasal humidified rapid-insufflation ventilatory exchange (THRIVE) is one of the high flow nasal cannula (HFNO) system. THRIVE is a product name [1].It differs from HFNO in that it provides 100% FiO2 as a heated and humidified oxygen flow. The HFNO system can provide different oxygen concentration (different FiO2). For non-intubated VATS, we have reported that THRIVE, rather than a nasal cannula, provides better oxygenation during one lung ventilation, and prevents dryness and heat loss from ventilation. [24,25]
Method (Page 3, line 106 to 110, material and methods, reference 24 and 25)
“In the non-intubated (N) group, patients were preoxygenated with Transnasal Humidified Rapid-Insufflation Ventilatory (THRIVE, Fisher & Paykel Healthcare, New Zealand) at an initial flow of 20 L/min before anesthesia. Oxygen flow was temporarily suspended immediately before iatrogenic pneumothorax and was resumed up to 10 L/min after the surgeon confirmed complete lung collapse using thoracoscopic observation [24,25].”
Reference:
[1] Patel,A.; Nouraei, SAR. Transnasal Humidified Rapid-Insufflation Ventilatory Exchange (THRIVE): a physiological method of increasing apnoea time in patients with difficult airways. Anaesthesia. 2015;70, 323-329.
[24]. Lai, CJ.; Yeh, KC.; Wang, ML.; Tai, WH.; Cheng, YJ. Heated humidified high-flow nasal oxygen prevents intraoperative body temperature decrease in non-intubated thoracoscopy. J Anesth. 2018, 32, 872-879.
[25]. Wang, ML.; Hung, MH.; Chen, JS.; Hsu, HH.; Cheng, YJ. Nasal high-flow oxygen therapy improves arterial oxygenation during one-lung ventilation in non-intubated thoracoscopic surgery. Eur J Cardiothorac Surg. 2018, 53, 1001-1006.
Point 14: p2L92 - rocuronium was used to facilitate tracheal intubation in the "I" group. This could influence the difference in recovery of swallowing. The authors should comment this in "Discussion" section.
Response 14: we thank the reviewer for their recommendation. To prevent residual rocuronium from affecting the swallowing recovery, all of our intubated patients were allowed to be transferred to the postoperative anesthetic care unit (PACU) after achieving a train-of-four (TOF) ratio greater than 1.0. We would add a paragraph on this to the Discussion (Page 11, line 328 to 330, discussion, reference 47):
“In our study, the neuromuscular blockade (NMB)-rocuronium was used during the operation. To avoid residual effect of the NMB, we used the train of four (TOF) monitor to confirm complete recovery from NMB. The criterion was a TOF ratio greater than 1.0 [47].”
Reference:
[47] Plaud, B.; Debaene, B.; Donati, F.; Marty, J. Residual paralysis after emergence from anesthesia. Anesthesiology. 2010, 112, 1013-1022.
Point 15: p3L94 - "sugammadex was administered after surgery with a TOF ratio 1:1 - what does this mean? - did they want to achieve TOFR of 1.0?
Response 15: The sugammadex administration followed the recommendation in Miller’s Anesthesia [29]. The details are as follows: a dose of 4 mg/kg is administered if spontaneous recovery of the twitch response has reached 1-2 post-tetanic counts and there are no twitch responses to TOF stimulation. A dose of 2 mg/kg is administered if spontaneous recovery has reached the reappearance of the second twitch in response to TOF stimulation [29]. The reversal of the displayed TOF ratio should be greater than 1, and then the patients can be extubated [30]. We have revised it as the follows (Page 3, line 133 to 135, material and methods, reference 29 and 30):
“The sugammadex administration followed the recommendation according to the response to TOF stimulation [29]. When the reversal of the displayed TOF ratio achieved a value greater than 1, the patients were extubated [30].”
Reference:
[29]. Murphy, G.; Eriksson, LI.; Miller, RD. Reversal (antagonism) of neuromuscular blockade. Miller's anesthesia, Ninth ed.; Gropper, M., Eriksson, L., Fleisher, L.; Wiener-Kronish, J., Cohen, N., Leslie, K.; Elsevier Health Science: 1600 John F. Kennedy Blvd. Ste 1600 Philadelphia, PA19103-2899, USA, 2019; Volume 1, pp. 853-854.
[30]. Suzuki, T.; Fukano, N.; Kitajima, O.; Saeki, S.; Ogawa, S. Normalization of acceleromyographic train-of-four ratio by baseline value for detecting residual neuromuscular block. Br J Anaesth. 2006, 96, 44-47.
Point 16: p3L129 - please explain this sentence - "We divided into two parts to analyze these..."
Response 16: we thanks the reviewer for their question. We have revised the statement as the follows (Page 4, line 171 to 172, material and methods):
“Bolus transmission through the esophagus and successful bolus transmission through the LES into the stomach were analyzed separately.”
Additionally, in previous studies, the experts also analyzed these processes separately [1-3], because their histology and anatomy of the areas are different.
Reference:
[1]. Lin, Z.; Imam, H.; Nicodeme, F.; Carlson, DA.; Lin, CY.; Yim, B.; Kahrilas, PJ. Flow time through esophagogastric junction derived during high-resolution impedance-manometry studies: a novel parameter for assessing esophageal bolus transit. Am J Physiol Gastrointest Liver Physiol. 2014, 307, 158-163.
[2]. Tutuian, R.; Vela, MF.; Shay, SS.; Castell, DO. Multichannel intraluminal impedance in esophageal function testing and gastroesophageal reflux monitoring. J Clin Gastroenterol. 2003, 37, 206-215.
[3]. Carlson, DA.; Omari, T.; Lin, Z.; Rommel, N.; Starkey, K.; Kahrilas, PJ.; Tack, J.; Pandolfino, JE. High-resolution impedance manometry parameters enhance the esophageal motility evaluation in non-obstructive dysphagia patients without a major Chicago Classification motility disorder. Neurogastroenterol Motil. 2017, 29, 10.1111/nmo.12941.
Point 17: p9L228 - "We did not experience any episode of hypoxia and reintubation in our study".
Response 17: we thanks the reviewer for their suggestion. We have revised the sentence in the following manner and hope this make it more clear (Page 7, line 234, result)
“We did not experience any episode of hypoxia and reintubation in our study.”
Point 18: p9L232 - ...are feasible and first apply...? what the authors do mean?
Response 18: we thanks the reviewer’s comment. We have rewritten it as the following (Page 10, line 281 to 284, discussion):
“Our results confirmed that effective oropharyngeal emptying is achieved immediately after complete emergence from general anesthesia, as proven by submental ultrasonography and high resolution impedance manometry (HRIM). This is the first study to apply submental ultrasonography and HRIM to assess effective postoperative swallowing.”

Reviewer 2 Report
Thank you for the opportunity to review this interesting manuscript on submental ultrasonography and high resolution impedance manometry for detection of oropharyngeal and esophageal emptying. The authors conducted a prospective randomized study comparing intubation vs non-intubation VATS. The study was registered and approved by the responsible research ethics committee. The paper is well written and organized properly. The methods are sound and reproducible. An exclusion flow-chart of study participants is available. The main results are presented focused and not too long. The discussion part includes a limitation section. The conclusions are supported by the results. I have only some minor points to be discussed.
Do the results support non-intubation VATS as standard of care?
The authors demonstrated only little differences regarding pharyngeal emptying and slightly higher differences in esophageal emptying. On the other hand, intubated patients were able to receive oral intake earlier compared to non-intubation patients. Any idea for the reasons?
BIS level and recovery times: Differences between N and I groups?
Could the results of this study have an impact on the question if and when patients should be allowed to drink in the recovery room?
Should there be further studies investigating on the latency of esophageal emptying after general anesthesia?
page 2 line 82: Did you mean 8 hours?
page 10 line 296: Please remove second full stop.
Thank you again and good luck with the revision.
Reviewer 3 Report
The authors describe a prospective trial to characterize swallowing after video-assisted thoracoscopic surgery with and without tracheal intubation.
Dysphagia is a common adverse effect during the postoperative period of patients who underwent tracheal intubation.
As impairment in swallowing and esophageal transport may lead to regurgitation and aspiration the study topic is of interest to the specialty. However, this study has limitations that limit the quality of the research and the conclusions which can be appropriately drawn from it. The following comments are offered for the authors considerations:
Major Considerations:
As the authors indicate, successful swallowing requires efficient oropharyngeal and esophageal emptying. The gold standard diagnostic for oropharyngeal dysphagia is videofluoroscopy. It is a reference standard for the diagnosis of oropharyngeal dysphagia to investigate temporal and spatial measurements of hyoid bone displacement during swallowing. Please refer to that in the introduction and the discussion.
I am not sure if I understood what the authors wanted to tell: should hyoid bone displacement (HBD) by submental ultrasonography and high resolution impedance manometry (HRIM) become standard of care after VATS? Or does the study just tell us, the difference of intubated versus non-intubated VATS is only minimal regarding postoperatively swallowing function?
How do these measurements add to the costs, and to staff time?
Introduction
LL 50: Previous studies indeed focus on ICU patients; but there are a lot more outcome parameters than subjective questionnaires. There are a multitude of studies in ICU patients regarding screening and assessment of dysphagia, the most often used diagnostic tool being flexible endoscopic evaluation of swallowing: e.g.
- Scheel R, Pisegna JM, McNally E, Noordzij JP, Langmore SE. Endoscopic assessment of swallowing after prolonged intubation in the ICU setting. Ann Otol Rhinol Laryngol. 2016
- A. Skoretz, H.L. Flowers, R. Martin:The incidence of dysphagia following endotracheal intubation: a systematic review.Chest, 137 (3) (2010), pp. 665-673
- B. Brodsky, J.L. Nollet, P.E. Spronk, M. González-Fernández: Prevalence, pathophysiology, diagnostic modalities and treatment options for dysphagia in critically ill patients [Published online ahead of print April 16, 2020] Am J Phys Med Rehabil (2020)
Methods
LL 94: what was the dosage of Sugammadex. Please specify in the methods and the results section.
LL 98: There is variability in measures of hyoid bone displacement and submental muscle size using the fixed or hand-held method of ultrasound transducer placement. Please specify your method and provide references for your decision.
Discussion
LL 232: Do you mean “were applied for the first time”
LL 243: Why do you think do patients who were tracheally intubated needed more time until they regain effective esophageal emptying? Please discuss this finding thoroughly.
May Sugammadex cause a reduction in esophageal motility? Please discuss that.
LL249: You state that HRIM is a powerful tool for detection of esophageal emptying lag. Do you recommend this tool on a regular basis in the PACU after VATS?
Conclusion
LL285: Please add “with complete recovery of neuromuscular function..”
LL 287: You state that there is little clinical effect by delayed esophageal emptying: How about regurgitation and aspiration?
LL 287: You state that HDM and HRIM are “warranted”. Please specify for what population and what kind of surgery.
Author Response
Response to Reviewer 3 Comments
Major Considerations:
Point 1: As the authors indicate, successful swallowing requires efficient oropharyngeal and esophageal emptying. The gold standard diagnostic for oropharyngeal dysphagia is videofluoroscopy. It is a reference standard for the diagnosis of oropharyngeal dysphagia to investigate temporal and spatial measurements of hyoid bone displacement during swallowing. Please refer to that in the introduction and the discussion.
I am not sure if I understood what the authors wanted to tell: should hyoid bone displacement (HBD) by submental ultrasonography and high resolution impedance manometry (HRIM) become standard of care after VATS? Or does the study just tell us, the difference of intubated versus non-intubated VATS is only minimal regarding postoperatively swallowing function?
How do these measurements add to the costs, and to staff time?

Response 1:
(1).We have added that video-fluoroscopy is a reference standard for the diagnosis of oropharyngeal dysphagia to investigate temporal and spatial measurements of hyoid bone displacement during swallowing in the introduction and discussion.
(a) Introduction (Page 2, line 53 to 59, introduction, reference 11 to 14): “In this study, muscle power as well as the fluid passage by sequential pressure transmission were used to investigate for both oropharyngeal and esophageal emptying. The gold standard diagnostic for oropharyngeal dysphagia is video-fluoroscopy [11]. Its spatial and temporal measurement of hyoid bone displacement (HBD) during swallowing has been widely applied [12,13]. However, the radiation exposure is still be concerned. HBD by submental ultrasonography, a non-invasive method without radiation exposure, was used to measure oropharyngeal muscle power. Its accuracy and reliability have been proven in comparisons with video-fluoroscopy [14].”
(b) Discussion (Page 10, line 284 to 286, discussion, reference 23 and 26): “Submental ultrasonography has been proven to have a good accuracy in HBD measurement when compared with video-fluoroscopy [23,26]. ”
(2). The aim of our study wanted to tell the readers that submental ultrasonography and HRIM are warranted for detecting postoperative oropharyngeal and esophageal emptying. These two measurements were seldom reported for surgical patients. However, besides anesthetic managements, some operations may affect the postoperative oropharyngeal emptying or esophageal emptying. HRIM could serve not only as a diagnostic method to detect postoperative esophageal dysfunction but also a useful tool of the follow-up.
We have revised in in the conclusion section:
(Page 11, line 355 to 361, conclusions): “The submental ultrasonography could be used in all surgeries that may affect patients’ oropharyngeal swallowing and for all patients with high aspiration risk, such as for older adults or stroke patients. It also helps to evaluate the rehabilitation of oropharyngeal swallowing after surgery. HRIM could be used in patients undergoing all surgeries associated with affected esophageal dysfunction, such as laparoscopic esophagectomy. HRIM could serve as a powerful tool for differential diagnosis on oral intake difficulties, and for postoperative follow-up.”
(3) HRIM measurements could also help doctors to identify the mechanisms of oral intake difficulties. Trained physicians in postoperative anesthetic care unit (PACU) could perform submental ultrasonography and HRIM at the same time when asking patients to swallow. The ultrasonography is popular in perioperative care areas where choking and aspiration are the main concern. As shown in our results, pre- and postoperative HBDs measurements could feasibly indicate oropharyngeal emptying to asses airway safety. We suggest measuring preoperative HBDs for patients with a history of choking or aspiration for postoperative comparison. The completeness of emergence was proven to play a major role in airway protection by allowing effective oropharyngeal emptying.
HRIM is rarely available in perioperative areas. A catheter is inserted through oropharyngeal, esophagus, lower esophageal sphincter into the stomach to evaluate esophageal emptying. Although it is well tolerated with limited discomfort, medical indications with patients’ consent and trained physicians are necessary to perform a successful evaluation. However, smooth oral intake is an important issue but not an emergency. The other examinations such as an endoscopy, barium swallow, pH meters could also be applied for postoperative dysphagia or difficult feeding. However, as HRIM is a good tool on esophageal function measured by both pressure and bolus transmission, sequential HRIM could be used to evaluate the postoperative recovery and the effectiveness of therapy.
Reference:
[11]Rommel, N.; Hamdy, S. Oropharyngeal dysphagia: manifestations and diagnosis. Nat Rev Gastroenterol Hepatol. 2016, 13, 49-59.
[12] Leonard, RJ.; Kendall, KA.; McKenzie, S.; Gonçalves, MI.; Walker, A. Structural displacements in normal swallowing: a videofluoroscopic study. Dysphagia. 2000, 15, 146-152.
[13] Perlman, AL.; VanDaele, DJ.; Otterbacher, MS. Quantitative assessment of hyoid bone displacement from video images during swallowing. J Speech Hear Res. 1995, 38, 579-585.
[14] Chen, YC.; Hsiao, MY.; Wang, YC.; Fu, CP.; Wang, TG. Reliability of Ultrasonography in Evaluating Hyoid Bone Movement. J Med Ultrasound. 2017, 25, 90-95.
[23] Hsiao, MY.; Chang, YC.; Chen, WS.; Chang, HY.; Wang, TG. Application of ultrasonography in assessing oropharyngeal dysphagia in stroke patients. Ultrasound Med Biol. 2012, 38, 1522-1528.
[26] Huang, YL.; Hsieh, SF.; Chang, YC.; Chen, HC.; Wang, TG. Ultrasonographic evaluation of hyoid-larynx approximation in dysphagic stroke patients. Ultrasound Med Biol. 2009, 35, 1103-1108.
Point 2: Introduction
LL 50: Previous studies indeed focus on ICU patients; but there are a lot more outcome parameters than subjective questionnaires. There are a multitude of studies in ICU patients regarding screening and assessment of dysphagia, the most often used diagnostic tool being flexible endoscopic evaluation of swallowing: e.g.
- Scheel R, Pisegna JM, McNally E, Noordzij JP, Langmore SE. Endoscopic assessment of swallowing after prolonged intubation in the ICU setting. Ann Otol Rhinol Laryngol. 2016
- A. Skoretz, H.L. Flowers, R. Martin:The incidence of dysphagia following endotracheal intubation: a systematic review.Chest, 137 (3) (2010), pp. 665-673
- B. Brodsky, J.L. Nollet, P.E. Spronk, M. González-Fernández: Prevalence, pathophysiology, diagnostic modalities and treatment options for dysphagia in critically ill patients [Published online ahead of print April 16, 2020] Am J Phys Med Rehabil (2020)
Response 2: thanks for the reviewer’s recommendation. We have added these articles in our reference and revised it as the following (Page 2, line 50 to 52, introduction, reference 6 to 10):
“Most previous studies focused on the effects of prolonged intubation (>48 hours) with questionnaires or flexible endoscopic evaluation of swallowing [6-10].”
Point 3: Methods
LL 94: what was the dosage of Sugammadex. Please specify in the methods and the results section.
Response 3: we thank the reviewer’s suggestion. The sugammadex administration followed the recommendation in Miller’s Anesthesia. The procedure can be described in detail as the following: a dose of 4 mg/kg is administered if spontaneous recovery of the twitch response has reached 1-2 post-tetanic counts and there are no twitch responses to train of four (TOF) stimulation. A dose of 2 mg/kg is administered if spontaneous recovery has reached the reappearance of the second twitch in response to TOF stimulation [29]. The reversal of the displayed TOF ratio should be greater than 1, and then the patients can be extubated [30].
We have revised it as the following in the method and result:
- (Page 3, line 133 to 135, material and methods, reference 29 and 30): “The sugammadex administration followed the recommendations according to eh response to TOF stimulation [29]. When the reversal of the displayed TOF ratio achieved a value greater than 1, the patients were extubated [30]”
- (Page 7, line 230 to 231, results):“In the I group, all patients were allowed to be transferred to the post-anesthetic care unit (PACU) after achieving a TOF ratio >1.”
Reference:
[29]. Murphy, G.; Eriksson, LI.; Miller, RD. Reversal (antagonism) of neuromuscular blockade. Miller's anesthesia, Ninth ed.; Gropper, M., Eriksson, L., Fleisher, L.; Wiener-Kronish, J., Cohen, N., Leslie, K.; Elsevier Health Science: 1600 John F. Kennedy Blvd. Ste 1600 Philadelphia, PA19103-2899, USA, 2019; Volume 1, pp. 853-854.
[30]. Suzuki, T.; Fukano, N.; Kitajima, O.; Saeki, S.; Ogawa, S. Normalization of acceleromyographic train-of-four ratio by baseline value for detecting residual neuromuscular block. Br J Anaesth. 2006, 96, 44-47.
Point 4: LL 98: There is variability in measures of hyoid bone displacement and submental muscle size using the fixed or hand-held method of ultrasound transducer placement. Please specify your method and provide references for your decision
Response 4: we thank the reviewer’s comment. Perry et al. showed that using the fixed method does not necessarily improve measurement accuracy of swallowing outcomes [22]. The fixed method uses mechanical head stabilization [2]. It impedes normal jaw movement, triggers compensatory muscle movement, and increases patients’ discomfort [22]. The hand held method is less invasive and more comfortable for participants [23]. We choose the hand held method.
We have revised the description and have added reference in follows (Page 3, line 98 to 99; 139 to 140, material and methods, reference 22 and 23):
(Line 98 to 99) “We measured HBD using submental ultrasonography before the surgery using the hand-held method [22,23]”..
(Line 139 to 140) “Postoperative HBD was immediately measured using submental ultrasonography [22,23].”
Reference:
[22] Perry, SE.; Winkelman, CJ.; Huckabee, ML. Variability in Ultrasound Measurement of Hyoid Bone Displacement and Submental Muscle Size Using 2 Methods of Data Acquisition. Folia Phoniatr Logop. 2016, 68, 205-210.
[2].Stone,M.; Davis, EP. A head and transducer support system for making ultrasound images of tongue/jaw movement. J Acoust Soc Am. 1995, 98, 3107-3112.
[23] Hsiao, MY.; Chang, YC.; Chen, WS.; Chang, HY.; Wang, TG. Application of ultrasonography in assessing oropharyngeal dysphagia in stroke patients. Ultrasound Med Biol. 2012, 38, 1522-1528.
Discussion
Point 5: LL 232: Do you mean “were applied for the first time”
Response 5: we thank the reviewer’s comment. We mean “ we are the first study to apply the methods for detecting effective postoperative oropharyngeal and esophageal changes” We have revised this sentence as the follows (Page 10, line 281 to 284, Discussion):
“Our results confirmed that effective oropharyngeal emptying is achieved immediately after complete emergence from general anesthesia, as proven by submental ultrasonography and high resolution impedance manometry (HRIM). This is the first study to apply submental ultrasonography and HRIM to assess effective postoperative swallowing.”
Point 6: LL 243: Why do you think do patients who were tracheally intubated needed more time until they regain effective esophageal emptying? Please discuss this finding thoroughly.
Response 6: we thank the reviewer’s comment. We have added our opinions as well as the cuff pressure description to the Discussion and Methods, respectively.
(1) Discussion (Page 10, line 297 to 301, Discussion, reference 1,38 and 39): “In our opinion, the cuff pressure of the endotracheal tube may affect upper esophageal motility [38,39], although the cuff pressure did not exceed 30 H2O in any of our patients. The motility of upper and lower portions of the esophagus influence each other [1]. This may be the cause of the delayed recovery of esophageal emptying.”
(2)Methods (Page 3, line 123 to 124, Material and Methods, reference 27):
“The cuff pressure of the single-lumen endotracheal tube (ST-ETT) was not allowed to exceed 30 cmH2O as measured by a pocket cuff pressure gauge [27].”
Reference:
[38] Rassameehiran, S.; Klomjit, S.; Mankongpaisarnrung, C.; Rakvit, A. Postextubation Dysphagia. Proc (Bayl Univ Med Cent). 2015,28, 18-20.
[39] Koo, CH.; Sohn, HM.; Choi, ES.; Choi, JY.; Oh, AY.; Jeon, YT.; Ryu, JH. The Effect of Adjustment of Endotracheal Tube Cuff Pressure during Scarless Remote Access Endoscopic and Robotic Thyroidectomy on Laryngo-Pharyngeal Complications: Prospective Randomized and Controlled Trial. J Clin Med. 2019, 8, 1787.
[1] Triadafilopoulos, G.; Hallstone, A.; Nelson-Abbott, H.; Bedinger, K. Oropharyngeal and esophageal interrelationships in patients with nonobstructive dysphagia. Dig Dis Sci. 1992, 37, 551-557.
[27] Lai, CJ.; Liu, CM.; Wu, CY.; Tsai, FF.; Tseng, PH. Fan, SZ. I-Gel is a suitable alternative to endotracheal tubes in the laparoscopic pneumoperitoneum and trendelenburg position. BMC Anesthesiol. 2017, 17, 3.
Point 7: May Sugammadex cause a reduction in esophageal motility? Please discuss that.
Response 7: Thanks for the reviewer’s comment. We added it in the following (Page 10, line 330 to 333, discussion, reference 48 to 50):
“Sugammadex, which selectively binds rocuronium and reverses its NMB action [48], does not have the same cholinergic effect as traditional anti-acetylcholinesterase [49], which was reported to affect esophageal tone [50]”
Reference:
[48]. Nicholson, WT.; Sprung, J.; Jankowski, CJ. Sugammadex: a novel agent for the reversal of neuromuscular blockade. Pharmacotherapy. 2007, 27, 1181-1188
[49]. Sasaki, N.; Meyer, MJ.; Malviya, SA.; Stanislaus, AB.; MacDonald, T.; Doran, ME.; Igumenshcheva, A.; Hoang, AH.; Eikermann, M. Effects of Neostigmine Reversal of Nondepolarizing Neuromuscular Blocking Agents on Postoperative Respiratory Outcomes: A Prospective Study. Anesthesiology. 2014, 121, 959-968.
[50] Kim, NY.; Koh, JC.; Lee, KY.; Kim, SS.; Hong, JH.; Nam, HJ.; Bai, SJ. Influence of reversal of neuromuscular blockade with sugammadex or neostigmine on postoperative quality of recovery following a single bolus dose of rocuronium: A prospective, randomized, double-blinded, controlled study. J Clin Anesth. 2019, 57, 97-102.
Point 8: L249: You state that HRIM is a powerful tool for detection of esophageal emptying lag. Do you recommend this tool on a regular basis in the PACU after VATS?
Response 8: HRIM is a powerful tool. However, it is rarely available in the PACU due to its high cost. However, for patients with a history of potential swallowing difficulties, HRIM could correctly identify the problems and confirm whether the patient has recovered from ineffective oropharyngeal and esophageal emptying. For uniportal VATS operations such as in this study, HRIM is not required to be routine because most patients displayed effective esophageal emptying within 100 minutes. However, for patients with a history of oral intake difficulties who underwent prolonged intubated VATS operations or thoracotomy, HRIM could be indicated perioperatively. In this study, we proved that esophageal emptying does not necessarily occur at the time of complete emergence after anesthesia and minimally invasive operations. Further investigation is needed to determine the effects of different types of anesthesia and operations on postoperative swallowing.
Conclusions
Point 9: LL285: Please add “with complete recovery of neuromuscular function..”
Response 9: we thank the reviewer’s suggestion. We have revised it as following in the manuscript
- (Page 11, line 346 to 350, limitation in the Discussion): “Fourth, our study used patients suitable for the non-intubated VATS for both non-intubated and tracheal intubated groups. Different respiratory patterns (spontaneous vs. controlled) may affect the postoperative swallowing. However, our results showed that complete emergence and the complete recovery of neuromuscular function ensure airway safety with effective oropharyngeal emptying.”
- (Page 11, line 353 to 355, conclusions): “Although effective oropharyngeal emptying without choking was proven to be achievable with complete emergence and complete recovery of neuromuscular function, anesthesia and tracheal intubation may delay effective esophageal emptying.”
Point 10: LL 287: You state that there is little clinical effect by delayed esophageal emptying: How about regurgitation and aspiration?
Response 10: we thank the reviewer’s opinion. The effective esophageal emptying is also important, because it means no regurgitant from stomach into the esophagus. Therefore, we removed the paragraph “little clinical effect” as the following (Page 11, line 353 to 355, conclusions):
“Although effective oropharyngeal emptying without choking was proven to be achievable with complete emergence and complete recovery of neuromuscular function, anesthesia and tracheal intubation may delay effective esophageal emptying.”
Point 11: LL 287: You state that HDM and HRIM are “warranted”. Please specify for what population and what kind of surgery.
Response 11: HRIM is a very powerful method for evaluating esophageal function. However, physicians only apply it in the internal medicine field or in the case of certain patients, such as those with achalasia, gastroesophageal reflux disease or after laparoscopic sleeve gastrectomy [1][2][3]. However, there are many more patients suffering from postoperative dysphagia. For example, maintaining or recovering from effective esophageal emptying is crucial for the success of operations and life quality of patients receiving head and neck cancer surgery or esophagectomy. Based on our experience, the results of questionnaires about oral intake difficulties and dysphagia in many patients after esophagectomy or spine surgery who presented normal level, are challenged by weight loss. The HRIM could help us identify the dysfunctional mechanisms as well as the outcomes after treatments.
We have rewritten the Conclusions section as the follow (Page 11, line 352 to 361, conclusions):
“Submental ultrasonography and HRIM are warranted for detecting postoperative oropharyngeal and esophageal emptying. Although effective oropharyngeal emptying without choking was proven to be achievable with complete emergence and complete recovery of neuromuscular function, anesthesia and tracheal intubation may delay effective esophageal emptying. Submental ultrasonography could be used in all surgeries that may affect patients’ oropharyngeal swallowing and for all patients with high aspiration risk, such as for older adults or stroke patients. It also helps to evaluate the rehabilitation of oropharyngeal swallowing after surgery. HRIM could be used in patients undergoing all surgeries associated with affected esophageal dysfunction, such as laparoscopic esophagectomy. HRIM could serve as a powerful tool for differential diagnosis on oral intake difficulties, and for postoperative follow-up.”
Reference:
[1]Lin, Z.; Carlson, DA.; Dykstra, K.; Sternbach, J. High-resolution Impedance Manometry Measurement of Bolus Flow Time in Achalasia and its Correlation with Dysphagia. Neurogastroenterol Motil. 2015, 27, 1232-1238.
[2] Ribolsi, M.; Holloway, R.; Emerenziani, S.; Balestrieri, P.; Cicala, M. Impedance-high resolution manometry analysis of patients with nonerosive reflux disease. Clin Gastroenterol Hepatol. 2014, 12,, 52-57.
[3]. Mion, F.;Tolone, S.;Garros, A.;Savarino,E.;Pelascini, E.;Robert, M.; Poncet, G.;Valette, PJ.; Marjoux, S.; Docimo, L.; Roman, S. High-resolution Impedance Manometry after Sleeve Gastrectomy: Increased Intragastric Pressure and Reflux are Frequent Events. Obes Surg. 2016, 26, 2449-2456.

Round 2
Reviewer 1 Report
This version of the manuscript is much more clear and significantly improved. The issues and queries have been sorted out mostly.
I have only minor comments:
Abstract:
Line 32: please provide statistical significance (p) for emptying at 100 min.
Material and Methods:
Line 88: ...class II obesity or higher... (Class III were not enrolled as well)
Line 96: The preoperative fasting time followed…
Line 104: the abbreviation „BIS“ should be explained
Results:
Line 233: All patients in both groups were sent to the PACU after restoration of BIS to the value of at least 85.
Reviewer 3 Report
Thanks for letting me review your revised manuscript. I am happy to see that the authors have tried to clarify most of the adressed issues.
I would recommend a native speaker for htorough editing of English language and style.
